# Influence of the Soluble–Insoluble Ratios of Cyclodextrins Polymers on the Viscoelastic Properties of Injectable Chitosan–Based Hydrogels for Biomedical Application

**DOI:** 10.3390/polym11020214

**Published:** 2019-01-26

**Authors:** Carla Palomino-Durand, Marco Lopez, Frédéric Cazaux, Bernard Martel, Nicolas Blanchemain, Feng Chai

**Affiliations:** 1Controlled Drug Delivery Systems and Biomaterials, University of Lille, Institut National de la Santé et de la Recherche Médicale (INSERM), Centre Hospitalier Régional Universitaire de Lille (CHU Lille), U1008, 59000 Lille, France; carla.palominodurand.etu@univ-lille.fr (C.P.-D.); marco.lopez@univ-lille.fr (M.L.); nicolas.blanchemain@univ-lille.fr (N.B.); 2UMET—Unité Matériaux et Transformations, University of Lille, Centre national de la Recherche Scientifique (CNRS), Institut National de la Recherche Agronomique (INRA), Ecole Nationale Supérieure de Chimie de Lille (ENSCL), Unité Matériaux et Transformations (UMR) 8207, 59655 Lille, France; frederic.cazaux@univ-lille.fr (F.C.); bernard.martel@univ-lille1.fr (B.M.)

**Keywords:** injectable hydrogels, water-soluble polymer of cyclodextrin, water-insoluble polymer of cyclodextrin, chitosan, shear-thinning, self-healing, viscoelastic properties

## Abstract

Injectable *pre-formed* physical hydrogels provide many advantages for biomedical applications. Polyelectrolyte complexes (PEC) formed between cationic chitosan (CHT) and anionic polymers of cyclodextrin (PCD) render a hydrogel of great interest. Given the difference between water-soluble (PCDs) and water-insoluble PCD (PCDi) in the extension of polymerization, the present study aims to explore their impact on the formation and properties of CHT/PCD hydrogel obtained from the variable ratios of PCDi and PCDs in the formulation. Hydrogels CHT/PCDi/PCDs at weight ratios of 3:0:3, 3:1.5:1.5, and 3:3:0 were elaborated in a double–syringe system. The chemical composition, microstructure, viscoelastic properties, injectability, and structural integrity of the hydrogels were investigated. The cytotoxicity of the hydrogel was also evaluated by indirect contact with pre-osteoblast cells. Despite having similar shear–thinning and self-healing behaviors, the three hydrogels showed a marked difference in their rheological characteristics, injectability, structural stability, etc., depending on their PCDi and PCDs contents. Among the three, all the best above-mentioned properties, in addition to a high cytocompatibility, were found in the hydrogel 3:1.5:1.5. For the first time, we gained a deeper understanding of the role of the PCDi/PCDs in the injectable *pre-formed* hydrogels (CHT/PCDi/PCDs), which could be further fine-tuned to enhance their performance in biomedical applications.

## 1. Introduction

Hydrogels consist of insoluble polymer networks swelled in water that present advantageous viscoelastic properties [1,2,3,4]. Hydrogels are very interesting in biomedical applications like tissue engineering and regenerative medicine (TERM) domains [2,4] as well as drug delivery [4] (such as proteins [3], growth factors, antibiotics, anti-inflammatories, etc. [2]) due to their unique characteristics such as biocompatibility, biodegradability, ability to mimic the extracellular matrix (ECM); and their capacity for controlled release of bioactive molecules [1,2,5]. Concerning the delivery to the injury area, injectable hydrogels have gained wider appreciation with their ease of handling [6], their capacity to fill irregular tissue defects of various sizes [7,8,9], and their potential for mini-invasive surgical procedures [2,7,10]. For designing effective injectable hydrogels, besides biocompatibility and biodegradability [2,11], their ability to be introduced inside the targeted site by injection from a syringe—a property referred to as an injectability—is a major requirement [3]. In this respect, injectable in situ forming hydrogels have exhibited many advantages, e.g., mechanical strength and sticking properties [12]. However, most in situ forming hydrogels are based on a chemical crosslinking reaction, e.g., using glutaraldehyde [13] or genipin during the gelation [14], which are time-sensitive and could diffuse some toxic components into the tissues and decrease the efficiency of hydrogel formation [3,15]. In contrast, injectable pre-formed physical hydrogels, thanks to their shear-thinning and self-healing properties, can overcome the above limitations [3,15].

Chitosan (CHT) has been extensively used in TERM [16,17,18] as an injectable material for repairing bone [19,20] and cartilage [8,10] due to its biocompatibility, bioactivity, and biodegradability [16,21,22]. Moreover, thanks to its polycationic character, CHT can interact spontaneously with anionic polymers in aqueous solutions to form polyelectrolyte complexes (PECs) by electrostatic interactions, thereby forming crosslinked networks without using any toxic chemical cross-linkers [1,23,24].

In such contexts, polymers of cyclodextrin (PCD) obtained from the crosslinking reaction of cyclodextrins (CD) with citric acid (CTR) [25] have gained our attention for their prolonged drug release properties and for their anionic character. In addition, we have reported that two forms of PCD, water-soluble (PCDs) and water-insoluble (PCDi) ones, could be collected from the reaction product [26]. During separation by filtration and dialysis, both PCDs and PCDi fractions presented the same chemical formula but differed in the extension of the polymerization reaction: PCDs displayed a hyperbranched globular structure (nanohydrogels) and is highly water-soluble, while PCDi was more extensively cross-linked and possessed a high swell ability in water [26,27].

Taking advantage of their opposite charges, our research team combined CHT and PCDs to form materials based on PECs for the elaboration of different sorts of drug delivery systems, such as a self-assembled layer-by-layer coating on textiles [28,29] and metallic vascular stents [30], or electrospun nanofiber membranes [31]. More recently, we also reported the feasibility of elaborating a spongy scaffold by freeze-drying CHT/PCDs hydrogels for wound dressing applications [32,33]; however, as an injectable hydrogel, it has not yet met the requirements for biomedical application. A study by Garcia–Fernandez et al. shed light on the different, but complementary properties of PCDi and PCDs, which were mixed and used as tablet excipients in variable ratios, and interestingly, influenced the disintegration time of the tablets in water [27]. Therefore, based on the hypothesis that such different properties of PCDs and PCDi would also impact the properties of CHT/PCD hydrogels, the present study investigated in depth the influence of the ratio of PCDi/PCDs on the formulation of CHT/PCD hydrogels on their shear-thinning and self-healing properties, injectability, cytocompatibility, etc. A deeper understanding of the role of the PCDi/PCDs ratio in the injectable pre-formed hydrogels would allow for fine-tuning in biomedical applications.

## 2. Materials and Methods

### 2.1. Materials

Chitosan (CHT, batch STBG1894V, *M_w_* = 190 kDa, determined by Gel permeation chromatography (GPC), degree of deacetylation = 73%, determined by ^1^H NMR), lactic acid (LA) and phosphate buffered saline (PBS, pH 7.4) were purchased from Sigma Aldrich (Lesquin, France). Water-soluble cyclodextrin polymer (PCDs) and water-insoluble cyclodextrin polymer (PCDi) were synthesized as reported by Martel et al. [26] and Garcia–Fernandez et al. [27]. Briefly, PCDs and PCDi were obtained from one single reaction of esterification between β–cyclodextrin (Kleptose^®^, a gift from Roquette, Lestrem, France) and citric acid (Sigma Aldrich, Lesquin, France) acting as a cross-linking agent and sodium hypophosphite as a catalyst. After the reaction, the water was removed in a Rotavapor (Büchi, Flawil, Switzerland), and the solid mixture was treated at 140 °C for 90 min under vacuum. Then the mixture was dispersed in water and filtered using a sintered glass funnel. The insoluble fraction (PCDi) was finally obtained after drying at 90 °C overnight, and the soluble fraction (PCDs) was further concentrated, purified by dialysis (Spectra/Por^®^, MWCO 20 kDa, Sigma Aldrich, Lesquin, France), and freeze-dried [26,27]. After the synthesis reaction, the yield of PCDs and PCDi fractions was 41% and 59%, respectively. The molar mass of PCDs was 20 kDa, determined by size exclusion chromatography with refractive index and dynamic light scattering detectors. The cyclodextrin weight percentage was determined by ^1^H NMR, composed of 58%, and it is estimated to be the same for both PCDs and PCDi. Ultrapure water (UPW = 18.2 MΩ·cm), which was used for all the experiments, was produced by the water purification system EGLA VEOLIA (Purelab flex, ELGA, High Wycombe, UK).

### 2.2. Methods

#### 2.2.1. Preparation of CHT/PCD Powders

The preparation of co-milled CHT/PCD powders was performed by adapting the protocol reported by Flores et al. [33]. Briefly, raw CHT, PCDs, and PCDi powders were milled and sieved at 125 µm. Subsequently, three types of powder mixtures were prepared by varying the weight ratio of CHT:PCDi:PCDs (3:0:3, 3:1.5:1.5 or 3:3:0), and co-milled using a mixer mill MM–40 (Retsch^®^, Haan, Germany) at 10 Hz for 3 min at room temperature (RT).

#### 2.2.2. Preparation of Injectable Hydrogels

The preparation of physical CHT/PCDi/PCDs hydrogels was carried out using a system of two interconnected syringes. Briefly, co-milled CHT:PCDi:PCDs powders were loaded in a first syringe and ultra-pure water (UPW) in a second one. Both syringes were connected via a Luer lock female-female connector (Vygon^®^, Ecouen, France). Powder and water components were then thoroughly mixed by pressing alternately on each of the plungers (about 80–90 repetitions) for 1 min. Subsequently, pure lactic acid (LA, >85%) was added into the mixture (for a final concentration of 1% *_v_*_/*v*_), and further mixed using the above-mentioned process for 1 min to obtain a hydrogel (Figure 1). The different formulations are shown in Table 1.

#### 2.2.3. Fourier Transform Infrared (FTIR) Spectroscopy Analysis

Hydrogels were freeze-dried in a freeze-dryer Alpha 1–2 (Christ^®^, Osterode am Harz, Germany) for 12 h (0.06 mBar, −53 °C), and characterized by FTIR spectroscopy using an Attenuated Total Reflectance ATR–FTIR Spectrum100 (Perkin–Elmer, Villebon-sur-Yvette, France). Sample spectra were recorded using a resolution of 4 cm^−1^ with 32 scans in a spectral range of 4000–650 cm^−1^.

#### 2.2.4. Scanning Electron Microscopy (SEM)

The microstructure of freeze-dried hydrogels was observed by using SEM (Hitachi S–4700, Düsseldorf, Germany) operating at an accelerating voltage of 5 kV and an emission current of 10 µA. Hydrogel disks were shaped in a 24-well plate and freeze-dried. Lastly, samples were sputter-coated with chrome before SEM observation.

#### 2.2.5. The Inverted-Vial Test for Gel Formation

The gelation behavior and stability of the hydrogels were evaluated using an inverted-vial test [34]. Immediately after mixing, equal quantities of each hydrogel were injected into a glass flask and incubated at 37 °C for 1 and 24 h, respectively. Finally, flasks were inverted, and gels images were registered.

#### 2.2.6. Rheological Analysis

In order to study the viscoelastic properties of hydrogels, rheological measurements were carried out by using an MCR 301 rheometer (Anton Paar, Les Ulis, France) with parallel plate geometry of 25 mm in diameter, a Peltier plate for temperature control, and a plate cover to prevent water evaporation during the analyses. A gap of 1 mm was fixed for all measurements, and hydrogels were loaded immediately after the mixing process in the syringes. The linear viscoelastic range (LVR) was determined in a strain sweep program at a constant frequency (1 Hz). The storage (G′) and loss modulus (G′′) of the hydrogels were evaluated as a function of time at a fixed temperature of 37 °C in the oscillatory mode at 1 Hz of frequency and 1% strain. Angular frequency sweep analyses were carried out at 1% strain from 1–100 rad/s at RT. Next, the shear-thinning and self-healing properties of the hydrogel pre-formed in the syringe were measured using the stress–strain method: Storage and loss modulus (G′ and G′′) were measured as a function of time, applying alternately the low shear condition (1% strain) for 3 min and the high shear condition (500% strain) for 2 min [15,35,36,37]. Five successive cycles of low/high shear conditions were applied at RT. Finally, in order to more specifically investigate the shear-thinning behavior of hydrogels, the flowability of the hydrogels was evaluated by performing a shear rate sweep (0–10^2^ s^−1^) program and measuring viscosity at RT. All tests were performed in triplicate.

#### 2.2.7. Injectability Test

The force applied to the syringe plunger to inject the hydrogels was measured by an Electromechanical Testing System (MTS Insight 100, Eden prairie, MN, USA) with a load cell of 100 kN. Briefly, 1.5 mL of each hydrogel was prepared by powder/liquid mixing, as mentioned above, in a 5 mL syringe (12 mm in diameter, Medicina^®^, Bolton, UK). Immediately after mixing, the syringe was fitted with an 18G syringe needle (1.2 mm in diameter × 40 mm in length, BD^®^ Microlance, Le Pont de Claix, France), and was placed over a sample vial filled with PBS (37 °C, pH 7.4). The load cell moved downwards until in contact with the plunger of the syringe. Thereafter, the compression force was applied at a crosshead speed of 10 mm/min on the syringe until the entire volume of hydrogel was ejected and the force–displacement curve was obtained. The morphology and structural integrity stability of the extruded hydrogel from the syringe, forming a cord directly immersed in PBS at 37 °C, was further observed. The images of hydrogel cords were taken after injection (AI) every 10 min during the first hour and subsequently after 24 h.

#### 2.2.8. Cytotoxicity Assay

The cytotoxicity of selected hydrogels was further evaluated using the extraction method (ISO 10993-5), with the pre-osteoblast MC3T3-E1 cell line (ATCC^®^ CRL-2594™, Manassas, VA, USA). Hydrogels were preconditioned for 24 h in serum-free Minimum Essential Medium (MEM–α, Gibco^®^, Thermo Fisher Scientific, Illkirch-Graffenstaden, France). Hydrogel extracts were prepared, for two time intervals (24 and 72 h) of extraction by adding 100 mg of hydrogel into 1 mL of MEM–α culture medium supplemented with 10% fetal bovine serum (FBS, Gibco^®^, Thermo Fisher Scientific, Illkirch-Graffenstaden, France) and incubated at 37 °C and with agitation at 80 rpm. The complete culture medium was also incubated under the same condition as the negative control. On the same day, MC3T3–E1 cells were plated at 4 × 10^3^ cells/well in a 96−well tissue culture polystyrenes plate and grown in 100 µL/well MEM–α medium supplement with 10% FBS at 37 °C, 5% CO_2_ for 24 h. The 96-well plate was partitioned into columns: Culture medium only (no cells); cells incubated in culture medium (control); and cells incubated in extraction medium. The medium for the monolayer cell culture was then replaced by the 100 μL/well sterile original hydrogel extracts (filtered through a 0.2 μm sterile syringe filter) or control medium. After 24 h of exposure of the cells to the hydrogel extracts or control at 37 °C, 5% CO_2_, the cell viability was determined using AlamarBlue^®^ (Gibco^®^, Thermo Fisher Scientific, Illkirch-Graffenstaden, France) assay. Briefly, the medium in each well was replaced with a 10% AlamarBlue^®^ solution in culture medium (200 μL/well), and the plate was incubated at 37 °C, 5% CO_2_ for 2 h. One hundred fifty microliters of reacted solution per well were transferred into a 96-well plate (Fluoro–LumiNunc^™^, ThermoScientific, Illkirch-Graffenstaden, France). The intensity of fluorescence was determined using a Twinkle LB 970 Microplate Fluorometer (Berthold, Bad Wildbad, Germany) with an excitation wavelength of 530 nm and an emission wavelength of 590 nm. The cell survival rate was expressed by the percentage of cell viability with respect to the value of the control. Each test was conducted in triplicate.

#### 2.2.9. Statistical Analysis

All quantitative data are presented as mean values ± SD. Statistical analysis of data was carried out using the software program Prism (version 5, GraphPad Software, La Jolla, CA, USA). The *t*-student test, one-way ANOVA and Tukey’s Post Hoc Test were also used. A *p*-value < 0.05 was determined as statistically significant.

## 3. Results and Discussion

### 3.1. Preparation of the Injectable CHT/PCDi/PCDs Hydrogel

All tested CHT:PCDi:PCDs ratios, upon mixing in the interconnected syringe system, immediately generated a transparent pale yellow gel, which had a homogeneous consistency without phase separation (Figure 1C). Briefly, the gelation process of the CHT/PCD hydrogel starts with the acidification of the aqueous suspension composed of the co-milled CHT:PCDi:PCDs powder. The acidification triggers the dissolution of the finely ground CHT particles and the protonation of its amino groups. Upon dissolution, solvated CHT chains (extended conformation) induce an increase in viscosity, allowing the available amino groups to interact with the carboxylate groups in the swollen PCDi particles and dissolved PCDs nanohydrogels respectively. It thereby generates a physical hydrogel crosslinked by a polyelectrolytes complex (PEC).

### 3.2. FT–IR Spectroscopy Analysis

In order to put in evidence of PEC formation, an FT–IR spectroscopy analysis of freeze-dried hydrogels was performed. In Figure 2A, the spectrum of the freeze-dried hydrogel 3:1.5:1.5 (CHT with both PCDi and PCDs) was compared with those of its principal components (CHT, PCDi, and PCDs). FT-IR spectra of CHT shows the characteristic stretching vibrations of –OH and –NH_2_ groups overlapping between 3600 and 3000 cm^−1^, and the C–H stretching vibrations between 3000 and 2700 cm^−1^. The other typical peaks for CHT are shown at 1650 cm^−1^ (C=O stretching of amide group), 1555 cm^−1^ (N–H bending vibrations of amide II), 1375.5 cm^−1^ (CH_3_ bending symmetric deformation), 1025.5 cm^−1^ (C–O stretching), and 894 cm^−1^ (C–O–C stretching) [33,38,39]. FT–IR spectra of PCDi and PCDs are similar due to their identical chemical compositions [26,27]. These spectra display peaks such as the typical –OH carboxylic acid vibration between 3600 and 3000 cm^−1^, also at 2931.5 cm^−1^ (CH_2_ asymmetric stretching), 1712.5 cm^−1^ (C=O stretching of carboxylic group), 1151 cm^−1^ (C–O–C stretching), and 1019.5 cm^−1^ (C–OH stretching), typical of sugar derivatives [28,40].

FT–IR spectra of the freeze-dried hydrogel shows the vibration band between 3600 and 3000 cm^−1^ and between 3000 and 2700 cm^−1^, which corresponds to the overlapping of the CHT and PCD signals respectively. The enlargement of the spectral region comprised between 2000 and 1200 cm^−1^ (Figure 2B) displays that, compared to that of the PCD spectra, the reduced peak height of C=O stretching at 1713 cm^−1^ (from carboxylic group) in the hydrogel spectrum could be associated with the conversion into carboxylate groups in the suspension [33]. Furthermore, the characteristic NH_2_ band of the CHT hydrogel (at 1555 cm^−1^) undergoes broadening and shifting to lower wavenumbers (1524.5 cm^−1^), which could be a consequence of intermolecular ionic interactions [41]. All these spectral changes in the hydrogel, resulting from the formation of a PEC between the NH_3_^+^ groups of CHT and the COO^−^ groups of PCD, have been also recorded by Flores et al. [33] for the hydrogel of 3:3:0.

### 3.3. Scanning Electron Microscopy (SEM)

The microstructure of hydrogels with different CHT:PCDi:PCDs ratios were analyzed using SEM in their freeze-dried form (Figure 3). The cross-sections of all hydrogels exhibited a similar interconnected porous structure (Figure 3A–C) with an average pore size of around 100 µm. A magnified view of all hydrogels (Figure 3A1–C1) showed that all of the walls of the pores are smooth and thin. Such a microstructure could potentially enable efficient nutrient exchange and waste removal that promotes cell growth and proliferation and is important for the scaffolds used in TERM [2]. However, no correlation was found between the hydrogel formulations and the porous structure. Moreover, according to the acquisition data (not shown) of high-resolution micro X-ray computed tomography, the porosities of all freeze-dried hydrogels were similar (around 87%).

### 3.4. Inverted-Vial Test for Gel Formation

The vial turn-over test demonstrates the gel formation as well as their stability. As shown in Figure 4, once the vials were turned upside down, none of the hydrogels flowed under the influence of gravity up to 24 h after being turned.

### 3.5. Rheological Analysis

The rheological evaluation of the viscoelastic properties (under stress) of hydrogels is shown in Figure 5. Time sweep analysis (Figure 5A) was performed within the linear viscoelastic range (LVR) at a frequency of 1 Hz and a strain of 1% at 37 °C in order to mimic body temperature. Amongst all ratios studied, that storage modulus (G′) was always greater than loss modulus (G′′) throughout the entire time interval. This means that the polyelectrolyte complex (PEC) formed a gel immediately after the injection of the hydrogel on the static plate of the rheometer. Anraku et al. [42] also demonstrated this phenomenon by mixing a 19% deacetylated chitin with an anionic sulfobutyl ether, β–cyclodextrin. The instantaneous formation of a tough and elastic hydrogel resulted from the electrostatic interactions between the amino and sulfate groups of both compounds respectively [42]. It should be highlighted that hydrogel 3:3:0 (only CHT and PCDi) showed a remarkably lower storage modulus (180 ± 10 Pa) compared to that (970 ± 50 Pa) of hydrogel 3:0:3 (only CHT and PCDs), while hydrogel 3:1.5:1.5 displayed an intermediate G′ value (498 ± 5 Pa). Nevertheless, all hydrogels (3:0:3, 3:1.5:1.5 and 3:3:0) presented similar loss modulus values: 135 ± 1 Pa, 109 ± 7 Pa, and 107 ± 3 Pa, respectively (Figure 5A).

The damping factor (tan δ), defined as the ratio G′′/G′, reflects the viscoelastic balance of the hydrogel [43]. The values of tan δ of the CHT/PCD hydrogels increased significantly (*p* < 0.05) as PCDi was added to the formulation, from 0.14 ± 0.01 for the gel containing only CHT and PCDs (ratio 3:0:3), to 0.22 ± 0.01 for hydrogel 3:1.5:1.5, and up to 0.61 ± 0.02 for the gel containing only CHT and PCDi (ratio 3:3:0). As reported by Borzacchiello and Ambrosio, chemical and physical hydrogels can be distinguished by their mechanical spectra, which are defined as the reliance between G′ and G′′ curves upon frequency. In general, a physical hydrogel shows a tan δ value higher than 0.1, which is typical of biological hydrogels based on polysaccharides or proteins [43]. The physical hydrogels can further be classified into, “weak” gels with higher tan δ values (reversible links formed from temporary associations between chains) and, “strong” gels with lower tan δ value (stable physical bonds formed in a given setting) [44]. According to this criterion, the hydrogel of ratio 3:3:0 is the weakest of the three, as it has the highest tan δ value (0.61 ± 0.02), which indicates a high loss of energy relative to the amount of energy stored.

Although PCDs and PCDi present identical chemical compositions and the same amount of COOH groups (i.e., 4 mmol of COOH groups per gram of PCD) [26], the rheological analysis of hydrogels showed that PCDs favors the CHT–PCD interaction through a higher intensity of PCD–COO^−^/CHT–NH_3_^+^ ionic force compared to PCDi. The reason would be that, in aqueous solution, the water-soluble hyperbranched globular structure of PCDs, observed by Herbois et al. [45], is about 50 nm in diameter, while the swollen insoluble PCDi particles are microscale, as it was ground and sieved on 125 µm mesh. Therefore, once dispersed in the hydrogels, PCDs presents a larger amount of available and accessible COOH groups in CHT/PCDs to form PEC with CHT than CHT/PCDi. This may explain the different viscoelastic properties of 3:3:0 and 3:0:3 hydrogels.

Figure 5B shows the evolution of G′ and G′′ as a function of angular frequency at 1% strain. For all three hydrogels, the modulus G′ was greater than G′′ throughout the analyzed frequency range, which confirms once again the elastic behavior of these hydrogels. The co-milled CHT/PCD powders, after having first been suspended in water, were followed by an acidification, thereafter a gel was formed inside the syringe. Upon the injection of this *pre-formed* gel, the phenomena of stress-induced gel–sol transition occurred due to the shear-thinning property of the gel. Subsequently, after extrusion from the syringe, the sol–gel transition occurred thanks to the self-healing properties of the hydrogel. In order to illustrate this gel–sol–gel transition in our gels, a rheologic study was performed using a continuous step-strain testing program, as shown in Figure 5C. A high strain (500%) was applied to the pre-formed hydrogels for 2 min to disrupt the ionic bond-formed network and to increase the flowability of the gels, as revealed by the drastic decrease of the G’ value. Thereafter, a low strain (1%) was applied, showing a sudden G’ jump, which was maintained for 3 min. In particular, the storage modulus G’ of hydrogel 3:0:3 recovered to ~72% of its initial value, which rose to ~90% after 3 min. In contrast, 3:3:0 and 3:1.5:1.5 hydrogels displayed similar G’ recovery in a lower range than that of the 3:0:3. It is noteworthy that such shear-thinning and self-healing properties shown under this step–strain program [35,36] were observed in a reproducible manner only for gels obtained from co-milled CHT/PCD powders and not for those formed from gently-blended CHT/PCD powders. This implies that the co-milling was a very important process that promoted close contact between CHT and PCD [33], which optimized the PEC formation. Indeed, this process also ensured the good homogeneity of the polymer network, in which electrostatic interactions could be reversibly dissociated and re-organized spontaneously, depending on the strain value applied.

Figure 5D reports the evolution of the viscosity of the function of increasing shear rate (0–10^3^ s^−1^). Viscosities of the three hydrogels rank in the order 3:0:3 > 3:1.5:1.5 > 3:3:0 within the shear rate domain investigated (26 s^−1^–1000 s^−1^). It was also found that the viscosity of all hydrogels decreased as the shear rate increased, e.g., the viscosity of the hydrogel of 3:1.5:1.5 decreased from 30 ± 1 Pa·s at 26 s^−1^ to 1.07 ± 0.04 Pa·s at 1000 s^−1^ (Figure 5D), demonstrating shear-thinning behavior [6]. This is a very important factor for an injectable pre-formed hydrogel, where the shear forces induced upon injection destroy the electrostatic interaction inside the hydrogel, decreasing the viscosity and allowing the hydrogel to flow through the needle and fill the target defect [15,35,46].

### 3.6. The Injectability Test

#### 3.6.1. Injection Force Profile

“Injectability” is the term used for defining the force required for activation of the syringe plunger for injection. This is an important parameter that determines the usability of the hydrogel by a clinician [15]. The force–displacement plots for the three hydrogel formulations are reported in Figure 6. The injection force firstly increased to about a 3 mm displacement, which then leveled out. Plateau values observed reflect the dynamic glide force, which were 30 ± 3 N, 26 ± 3 N, and 17 ± 1 N for the hydrogels 3:0:3, 3:1.5:1.5, and 3:3:0, respectively. Thus, the injection forces of the three gels ranked in the same order as their viscosity values reported in the rheological analysis.

The injection force of fluid material is influenced by the inner diameters of the syringe and needle, as well as the crosshead speed [15,47,48]. Thus, the measurement of injectability force often varies in the published literature. Nevertheless, the study of Cilurzo et al. pointed out that an injection force lower than 125 mPa (45 N) represents an easy injection for the physician [49,50]. In such a context, all tested hydrogels displayed a maximum force value below 45 N, which ensures an injection without difficulty for the clinicians.

#### 3.6.2. In Batch Structural Integrity of Hydrogels

In order to evaluate the stability of the hydrogels after injection in an aqueous environment, all samples were extruded through an 18G needle directly into PBS at pH 7.4 at 37 °C. All three injected hydrogels showed an unbroken cord-like structure with a smooth surface (Figure 7, after injection—AI). However, the hydrogel with the 3:3:0 ratio (only CHT and PCDi) started to deform and to disintegrate within 10 min. This may be due to a high swelling rate (10 times its initial volume) of the PCDi [26], which could provoke a strong swelling of the hydrogel causing the disintegration of the cord. On the contrary, the extruded cord from hydrogel 3:0:3 (only CHT and PCDs) showed a visible decrease in diameter (shrinking) after 30 min of immersion in PBS. Despite this feature, good cohesion and integrity were maintained in the PBS for up to 24 h. Interestingly, 1 h after its injection in PBS, the cord of hydrogel 3:1.5:1.5 did not show any change in terms of volume or integrity (Figure 7, 1 h). Apart from showing a rougher surface, the hydrogel composed of both PCDi and PCDs displayed the best dimensional stability and structural integrity even 24 h after immersion in PBS (Figure 7, 24 h), compared to the other two gels, which has been confirmed by observation of longer incubation up to 7 days (shown in Appendix A). Therefore, the combination of both water-soluble and water-insoluble PCD in the formulation of the CHT/PCD hydrogel shows the most desirable characteristics without destabilizing the formed ionic network. Additionally, all hydrogels were injected into an empty flask before adding the PBS (Figure 8) and were incubated for 24 h. The images show the evolution of the gels in aqueous medium: The hydrogel 3:0:3 displayed a decrease in its volume in contrast to hydrogel 3:3:0, which drastically swelled and so increased in volume. On the other hand, the shape of hydrogel 3:1.5:1.5 was not affected during the incubation; however, its volume increased slightly (Figure 8, 24 h). These observations are consistent with those shown above (Figure 7).

Rapid gel–sol–gel transition of the hydrogel during the extrusion from the syringe and rapid self-healing behavior, in addition to dimensional stability, are the most important parameters that would ensure a perfect filling of a tissue defect [15,51], e.g., bone defects [52], by injection. Moreover, the absence of shrinking of the hydrogel would ensure good contact of the injected material with the wall defect and good integrity for the necessarily long period of time required for the in situ therapeutic activity (i.e., drug release or delivery of cells) [3]. Finally, considering viscoelastic properties (shear-thinning and self-healing), injection force, and structural integrity after extrusion from the syringe, the hydrogels with formulations 3:0:3 and 3:1.5:1.5 best responded to the specifications of the application and were further evaluated in the following cytotoxicity tests.

### 3.7. Cytotoxicity Assay

The cell response to hydrogels is essential for biomedical applications. Since the ultimate objective of this study is to elaborate a bioactive hydrogel for bone regeneration, we chose the pre-osteoblast MC3T3–E1 cell line for investigating the cytotoxicity along with bioactivity (i.e., the bone cell reaction to our studied materials). The cytotoxicity of two selected hydrogels was evaluated by determining the cell viability and is shown in Figure 9. The cell survival rate (cell viability with respect to the control) in the 24-hour extract medium of hydrogels at ratios 3:0:3 and 3:1.5:1.5 were 92 ± 4% and 90 ± 7%, respectively. No significant difference was found between groups (*p* > 0.05). Moreover, the cell survival rate in the 72-hour extract medium is similar to that of the 24-hour ones, and there is no significant difference between groups either (*p* > 0.05). Based on these results, hydrogels with ratios 3:0:3 and 3:1.5:1.5 are non-cytotoxic, and will be considered in future investigations. The optimized hydrogel will be further evaluated for directly incorporating cells or bioactive growth factors to promote bone repair and regeneration.

## 4. Conclusions

In this study, CHT/PCD-based physical hydrogels were developed from water-soluble and/or water-insoluble forms of citric acid–β–cyclodextrin polymer. The rheological study showed that polyelectrolyte complex formation between cationic CHT and anionic PCD endowed the hydrogels with shear-thinning and self-healing properties. Moreover, it clearly highlighted that the influence of the ratio of the PCDi and PCDs used plays an essential role on the hydrogel’s properties due to their different natures detailed throughout this paper. A gel based on equivalent amounts of PCDs and PCDi could be a good compromise to both reduce shrinkage/swelling behavior (hence, better dimensional stability) and enable a high cytocompatibility towards pre-osteoblastic cells. Concerning the future developments, the addition of calcium phosphate (e.g., hydroxyapatite) to this hydrogel could bring further improvements to the structural support (osteoconductivity), cell delivery, and/or bioactive molecules release (e.g., growth factors) to the material in order to create an appropriate environment for cell adhesion, migration, proliferation, and differentiation in bone tissue engineering applications.

## Figures and Tables

**Figure 1 polymers-11-00214-f001:**
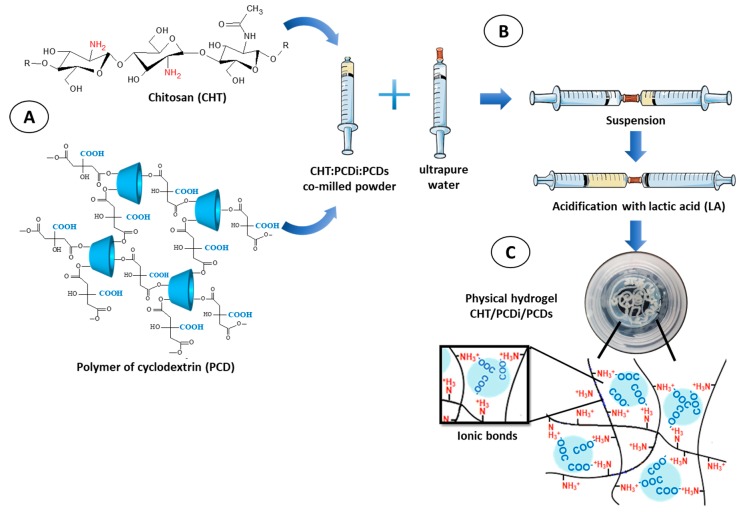
Schematic representation of the formation of CHT/PCDi/PCDs hydrogels. (**A**) The chemical structure of CHT and PCD, i.e., the co-milled powders loaded in one of the syringes; (**B**) the double-syringe system used for the preparation of hydrogel: The suspension phase, followed by the acidification to generate the PEC; (**C**) the macroscopic appearance of an injected CHT/PCDi/PCDs physical hydrogel (at a ratio of 3:1.5:1.5).

**Figure 2 polymers-11-00214-f002:**
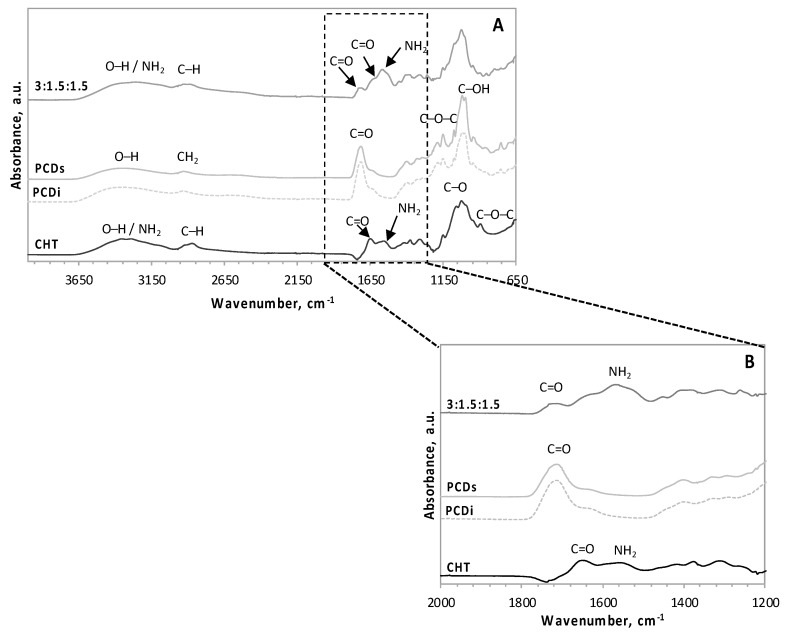
FT–IR spectra of (**A**) the CHT:PCDi:PCDs hydrogel with a 3:1.5:1.5 ratio, the raw CHT, PCDi, and the PCDs spectra, and (**B**) the magnification of the wavenumber region between 2000 and 1200 cm^−1^.

**Figure 3 polymers-11-00214-f003:**
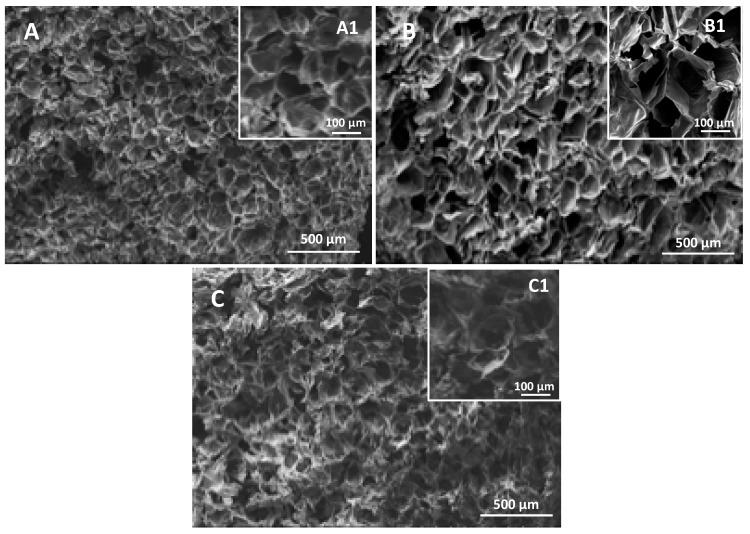
SEM micrographs of cross-sections of freeze-dried hydrogels CHT:PCDi:PCDs: (**A**,**A1**) 3:0:3, (**B**,**B1**) 3:1.5:1.5 and (**C**,**C1**) 3:3:0. Images (A–C) are taken at ×50 magnification and (**A1**–**C1**) are taken at ×100 magnification.

**Figure 4 polymers-11-00214-f004:**
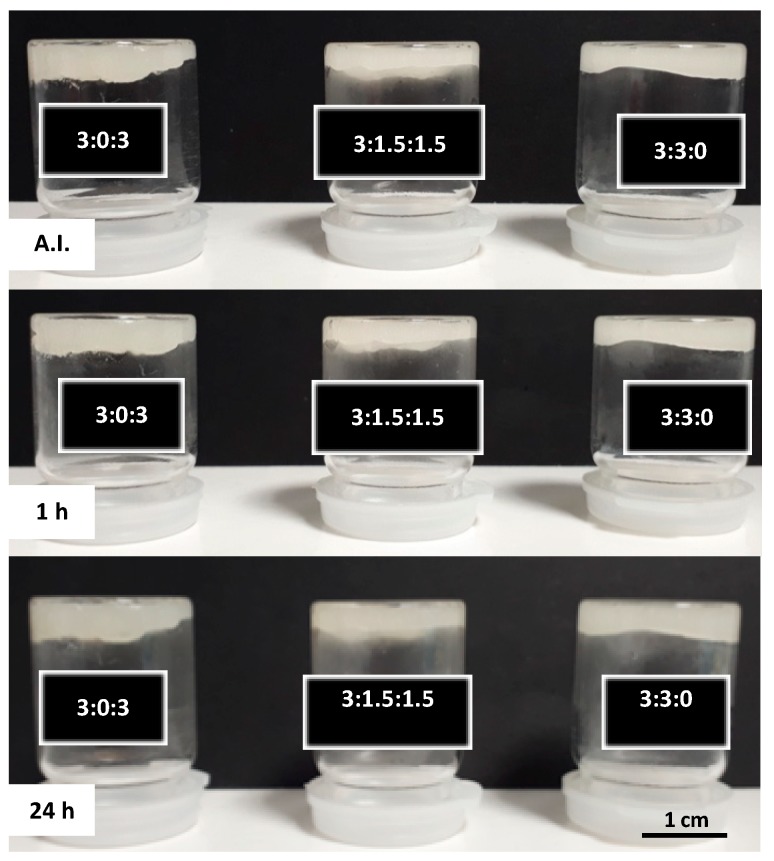
Observation of inverted vials containing hydrogels with different CHT:PCDi:PCDs ratios immediately after injection (AI) at room temperature (RT) and after 1 h and 24 h of incubation at 37 °C.

**Figure 5 polymers-11-00214-f005:**
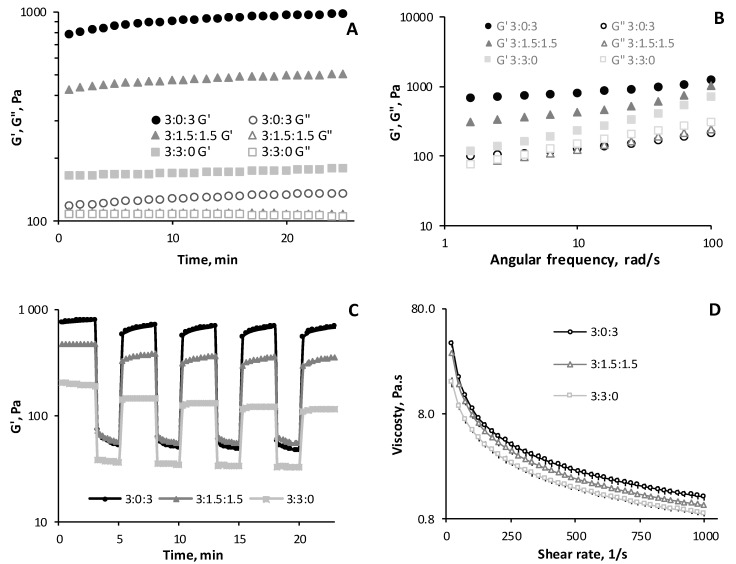
Rheological analysis of the hydrogels at different CHT:PCDi:PCDs ratios: (**A**) Evolution of the storage (G′) and loss (G′′) modulus as a function of time at a frequency of 1 Hz at 37 °C (the G′′ curve of hydrogel 3:1.5:1.5 is overlapped by the G′′ curve of hydrogel 3:3:0). (**B**) the frequency sweep determining G′ and G′′ at a constant strain of 1%. (**C**) The self-healing evaluation by cyclic step–strain measurements at a low strain of 1% and at a high strain of 500% at 1 Hz of frequency at RT, and (**D**) the shear-thinning evaluation by shear-rate dependent variations of viscosity at RT. Average values obtained from three different measurements are represented in all graphs.

**Figure 6 polymers-11-00214-f006:**
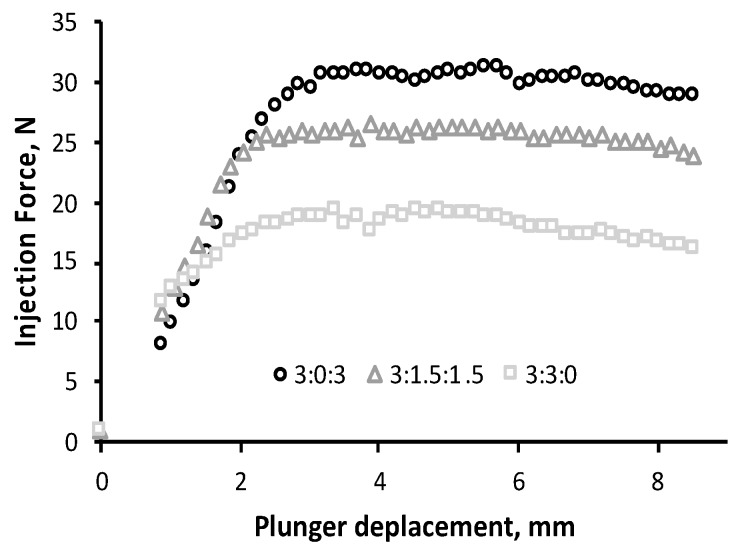
Injection force profile using a 5–mL syringe (12 mm of inner diameter) with an 18G needle of CHT:PCDi:PCDs hydrogels at a crosshead speed of 10 mm/min at room temperature. The curves correspond to the average values of three experiments.

**Figure 7 polymers-11-00214-f007:**
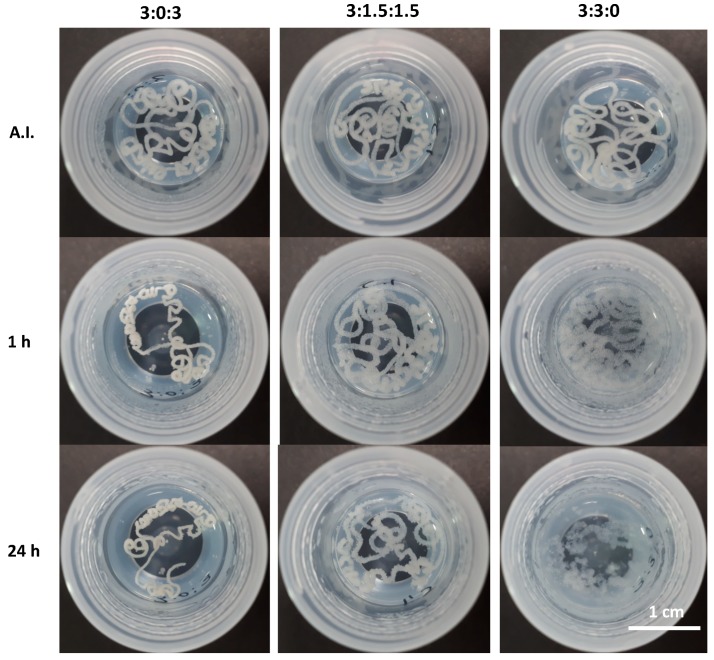
Qualitative evaluation of the integrity and self-healing properties of CHT:PCDi:PCDs hydrogels after injection with an 18G needle in PBS at 37 °C: Immediately after injection (AI) and after 1 h and 24 h of incubation.

**Figure 8 polymers-11-00214-f008:**
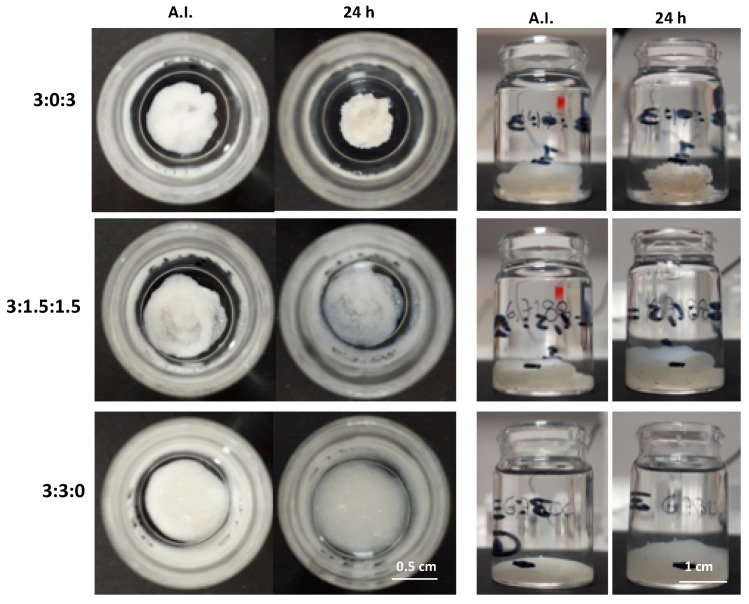
Evolution of CHT:PCDi:PCDs hydrogels in PBS at 37 °C: Immediately after injection (AI) and after 24 h of incubation.

**Figure 9 polymers-11-00214-f009:**
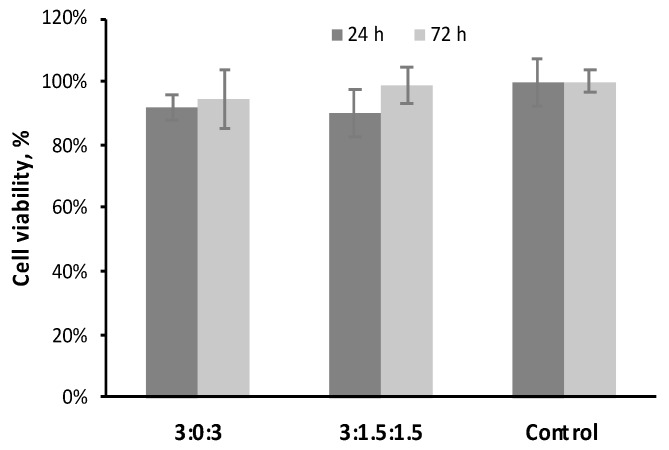
Cell viability of pre-osteoblast cells (MC3T3–E1) after 24 h of exposure to the 24-hour and 72-hour extracts of hydrogels at ratios 3:0:3 and 3:1.5:1.5. MEM–α culture medium was used as a control. No statistically significant difference was found between any hydrogel and control (*p* > 0.05).

**Table 1 polymers-11-00214-t001:** Formulations for preparing the CHT/PCDi/PCDs hydrogels.

Group	CHT, % _*w*/*v*_	PCDi, % _*w*/*v*_	PCDs, % _*w*/*v*_	UPW, % _*v*/*v*_	LA, % _*v*/*v*_
3:0:3	3	0	3	93	1
3:1.5:1.5	3	1.5	1.5	93	1
3:3:0	3	3	0	93	1

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
