# Peer review of "Influence of the Soluble–Insoluble Ratios of Cyclodextrins Polymers on the Viscoelastic Properties of Injectable Chitosan–Based Hydrogels for Biomedical Application"

_polymers, 2019, doi:10.3390/polym11020214_

Round 1

Reviewer 1 Report

The manuscript describes the properties of a novel family of shear-thinning and self-healing hydrogels prepared by mixing chitosan with soluble and insoluble citric acid-bonded β-cyclodextrin polymers. The authors highlighted the ternary formulation that ensures the optimal viscoelastic properties for potential biomedical applications. Microstructure, injectability, structural integrity and cytotoxicity were investigated as well.

The experimental work is solid, the results are consistent and well presented. The manuscript can be accepted for publication on this journal, after minor revision. The following points should be addressed.

1- What is the amount/yield of soluble and insoluble fractions, obtained from the synthesis reaction?

2- Line 93-94. Both soluble and insoluble fractions have the same content of cyclodextrin (i.e. 58 %)?

3- Line 240. Please, correct “demonstratsthe”.

4- Line 271. According to what is reported in lines 265-271, should not the hydrogel of ratio 3:3:0 be the strongest one, since it has the highest tan δ value?

Author Response

1-What is the amount/yield of soluble and insoluble fractions, obtained from the synthesis reaction?

Response 1: After the synthesis reaction of PCD, the yield of soluble and insoluble fractions was 41% and 59%, respectively. This information was properly added in the article (Line 93-94 of revision version).

2- Line 93-94. Both soluble and insoluble fractions have the same content of cyclodextrin (i.e. 58 %)?

Response 2: In fact, as described in manuscript, PCDs and PCDi are soluble and insoluble fractions obtained from one single polymerization experiment, starting from a mixture of CTR/catalyst/CD of weight ratio 10:3:10. After reaction, both fractions are separated by filtration on a sintered glass. As previously reported by Martel et al [ref. XX, Line 96 of revision version], the reaction provokes the formation of cyclodextrin citrate that progressively polymerize as function of time. For short time of polymerization, only PCDs is obtained, but the reaction yield is low. Therefore, the optimal yield in PCDs is obtained if the reaction time is prolonged, but also resulting in the formation of PCDi. So PCDi is obtained by the increase of the extension of the polymerization of PCDs. Overall, the difference between PCDs and PCDi corresponds to their respective degrees of polymerization (forming respectively hyperbranched/crosslinked macromolecular structures), rather than to their CD content. The CD content can be analysed qualitatively by proton NMR, but cannot be quantified by this technique, as PCDi forms a hydrogel in D2O. To conclude, the CD content in PCDi is estimated to be the same as that of PCDs. The same approach is considered for epichlorohydrin based CD polymers.

3- Line 240. Please, correct “demonstratsthe”.

Response 3: Sorry for typos. This typo has been corrected into “demonstrate the”. Our manuscript was once again proof-read by our mother-tongue colleague and ourselves (shown by different tracking colors), the corresponding corrections were made in revised version of manuscript.

4- Line 271. According to what is reported in lines 265-271, should not the hydrogel of ratio 3:3:0 be the strongest one, since it has the highest tan δ value?

Response 4: Sorry for the confusion, there is the error of description in line 269-270, the correct one is “The physical hydrogels can further be classified into “weak” gels with lower higher tan δ value (reversible links formed from temporary associations between chains) and “strong” gels with higher lower tan δ value (stable physical bonds formed in a given setting)”; thus, the hydrogel of ratio 3:3:0 was the weakest one.

Reviewer 2 Report

The manuscript entitled " Influence of the solubleinsoluble ratios of cyclodextrins polymers on the viscoelastic properties of injectable chitosanbased hydrogels for biomedical application" by Carla Palomino-Durand et al. reports on elaboration and use of injectable chitosan-based hydrogels with variable ratio of cyclodextrins polymers for biomedical applications caused by different viscoelastic properties. The manuscript is well structured, suitable for Polymers and intersting for its readers. It is written in an acceptable English but it still requires several corrections\improvements\changes to be ready for publication in Polymers. They are listed below, authors should taken into account for revision and resubmission

1) Few typos should be corrected by a moderate English revision through mother tongue revisor

2) Few novel reviews\articles should be added by updating Literature references

3) In the cytotoxicity studies 48h are missing and growing time different from 24 hours seems to be  ovelooked. Please add discussion about this point

4) What was the reason for cell choice? Please comment further

5) Figure 8: what about longer incubation? Commen\explain

6) What was time stability of hydogels? What about growth factors loadiing\release\uptake mechanism?

7) A more dedicated sentences remarking specific biomedical applications in the discussion\outlook could improve the comprehesion of manuscript.

8) The porosity of hydogels should be quantified. Please provide data\discuss further

Author Response

1) Few typos should be corrected by a moderate English revision through mother tongue reviser

Response 1: Sorry for typos. Our manuscript was once again proof-read by our mother tongue colleague and ourselves (shown by different tracking colors), the corresponding corrections were made in revised version of manuscript.

2) Few novel reviews\articles should be added by updating Literature references

Response 2: we have updated reference by adding following the most recent reviews\articles:

4Joshi, S.; Vig, K.; Singh, S.R. Advanced Hydrogels for Biomedical Applications. 2018, 5, 5

17. Pellá, M.C.G.; Lima-Tenório, M.K.; Tenório-Neto, E.T.; Guilherme, M.R.; Muniz, E.C.; Rubira, A.F. Chitosan-based hydrogels: From preparation to biomedical applications. Carbohydrate Polymers 2018, 196, 233–245

18. Shariatinia, Z.; Jalali, A.M. Chitosan-based hydrogels: Preparation, properties and applications. International Journal of Biological Macromolecules 2018, 115, 194–220    

37. Chen, H.; Cheng, J.; Ran, L.; Yu, K.; Lu, B.; Lan, G.; Dai, F.; Lu, F. An injectable self-healing hydrogel with adhesive and antibacterial properties effectively promotes wound healing. Carbohydrate Polymers 2018, 201, 522–531

51. Alarçin, E.; Lee, T.Y.; Karuthedom, S.; Mohammadi, M.; Brennan, M.A.; Lee, D.H.; Marrella, A.; Zhang, J.; Syla, D.; Zhang, Y.S.; et al. Injectable shear-thinning hydrogels for delivering osteogenic and angiogenic cells and growth factors. Biomater. Sci. 2018, 6, 1604–1615

 3) In the cytotoxicity studies 48h are missing and growing time different from 24 hours seems to be overlooked. Please add discussion about this point

Response 3: For our presented cytotoxicity studies, we followed strictly required procedures for elution test method by ISO 10993-5:

The extraction duration for a device was specified as 24 hours (at 37°C) for the medical devices that are in short-term contact (no longer than 30 days) with body; or prolonged duration i.e. 72 hours (at 37°C) for the medical devices that are in long-term contact (longer than 30 days) with body. Therefore, in this study we conducted two extraction durations.

The exposure duration of cultured cells in contact with extracts of a device was specified as 24 hours (37°C / 5% CO2). Therefore, we evaluated the cell response to extract of hydrogel after 24-hour exposure.

To make it clearer, we revised the description of method of “cytotoxicity assay” for better discriminating extraction duration and cell exposure duration.

4) What was the reason for cell choice? Please comment further

Response 4: According to ISO 10993-5, established cell lines are preferred. In the present study, pre–osteoblast MC3T3–E1 cell line (ATCC® CRL–2594™, USA) was chosen for evaluating the cytotoxicity along with a bioactivity. Since the ultimate objective of this study is to elaborate a composite active hydrogel (CHT/PCD/Hydroxyapatite) for bone regeneration, we chose pre-osteoblast cell line for investigating the bone cell reaction to our studied materials.

We have also added this reason of cell choice and comment in the manuscript with revision tracking (between Line 399 and 402).

5) Figure 8: what about longer incubation? Comment\explain

Response 5: In fact, for both Fig 7 and Fig 8, the observation the stability of hydrogels in aqueous environment was conducted up to 7 days, for not loading the manuscript with so many pictures, we selected the representative durations: 1 h and 24 h. While we can supply whole sum of pictures of “long-term” observation up to 7 days as supplement information (except some images missing in Fig 8 for 7 days). As we can see, when incubating up to 7 days, the cord of hydrogel 3:1.5:1.5 did not show much change in terms of volume or integrity comparing to that of 24-hour immersion. While the hydrogel at ratio 3:3:0 (only CHT and PCDi), deformed and disintegrated even more after 7-day incubation than that of 24-hour immersion; the hydrogel 3:0:3 (only CHT and PCDs) showed more decrease in diameter (shrinking) after 7 day of immersion in PBS than 24-hour one. We added this supplement information in the results (between Line 371 and 372).

6) What was time stability of hydrogels? What about growth factors loading\release\uptake mechanism?

Response 6: As mentioned above, the studied hydrogel has rather good time stability, thus, the release of incorporated growth factor cannot rely on the erosion of platform. Strategies for growth factor incorporation into scaffolds will be non-covalent (either surface adsorption, physical entrapment, affinity binding to PCD or ionic complexation), which has much less impact on the physicochemical properties and interactions of the growth factor than chemical conjugation. The release of growth factor from hydrogel will depend on the degree of swelling and diffusion.

7) A more dedicated sentences remarking specific biomedical applications in the discussion\outlook could improve the comprehension of manuscript.

Response 7: Yes, the specific biomedical application for this elaborated hydrogel is for bone tissue regeneration. This is clarified in discussion (Line 409) and restressed in the conclusion/outlook (Line 428) in revised version of manuscript.

8) The porosity of hydrogels should be quantified. Please provide data\discuss further

Response 8: According to the acquisition data (not shown for space) of high-resolution micro X-ray computed tomography, the porosity of all freeze-dried hydrogels was similar (around 87%). This is also added in Line 245-246.

Round 2

Reviewer 2 Report

Authors have addressed satisfactorily all issues raised by previous review and in current revised form can be accepted for publication in Polymers